# Enhancing the Effects of Neurofeedback Training: The Motivational Value of the Reinforcers

**DOI:** 10.3390/brainsci11040457

**Published:** 2021-04-03

**Authors:** Rubén Pérez-Elvira, Javier Oltra-Cucarella, José Antonio Carrobles, Jorge Moltó, Mercedes Flórez, Salvador Parra, María Agudo, Clara Saez, Sergio Guarino, Raluca Maria Costea, Bogdan Neamtu

**Affiliations:** 1Neuropsychophysiology Laboratory, NEPSA Rehabilitación Neurológica, 3003 Salamanca, Spain; rubenperezelvira@gmail.com (R.P.-E.); mjagudojuan@gmail.com (M.A.); clarasaezbueno@gmail.com (C.S.); 2Department of Health Psychology, Universidad Miguel Hernández de Elche, 03202 Elche, Spain; 3Biological and Health Psychology Department, Universidad Autónoma de Madrid, 28049 Madrid, Spain; joseantonio.carrobles@uam.es; 4PSYD-Neurofeedback, 46022 Valencia, Spain; trinyjorge@hotmail.com (J.M.); mercedesfbps@outlook.es (M.F.); 5LURIA Rehabilitación Neurológica, 11407 Jerez, Spain; salvador@luriajerez.es; 6NEPSA Rehabilitación Neurológica, 47001 Valladolid, Spain; sergio.guarino.87@gmail.com; 7Research Department (Ceforaten), Sibiu Pediatric Hospital, 550178 Sibiu, Romania; ralucacosteadr@gmail.com (R.M.C.); bogdan.neamtu@ulbsibiu.ro (B.N.); 8Faculty of Medicine Lucian Blaga, University from Sibiu, 550169 Sibiu, Romania; 9Faculty of Engineering, Lucian Blaga, University from Sibiu, 550025 Sibiu, Romania

**Keywords:** neurofeedback, reinforcer, sensorimotor rhythm

## Abstract

The brain activity that is measured by electroencephalography (EEG) can be modified through operant conditioning, specifically using neurofeedback (NF). NF has been applied to several disorders claiming that a change in the erratic brain activity would be accompanied by a reduction of the symptoms. However, the expected results are not always achieved. Some authors have suggested that the lack of an adequate response may be due to an incorrect application of the operant conditioning principles. A key factor in operant conditioning is the use of reinforcers and their value in modifying behavior, something that is not always sufficiently taken into account. This work aims to clarify the relevance of the motivational value versus the purely informational value of the reinforcer. In this study, 113 subjects were randomly assigned two different reinforcer conditions: a selected reinforcer—the subjects subjectively selected the reinforcers—or an imposed reinforcer—the reinforcers were assigned by the experimenter—and both groups undertook NF sessions to enhance the sensorimotor rhythm (SMR). In addition, the selected reinforcer group was divided into two subgroups: one receiving real NF and the other one sham NF. There were no significant differences between the groups at baseline in terms of SMR amplitude. After the intervention, only those subjects belonging to the selected reinforcer group and receiving real NF increased their SMR. Our results provide evidence for the importance of the motivational value of the reinforcer in Neurofeedback success.

## 1. Introduction

The field of neuroscience has experienced a considerable growth in recent years [1]. The ultimate goal of the disciplines included under the neuroscience term is the brain function analysis and understanding, in health and disease, to discover the best therapies for brain disorders [2]. Neuroscience conveys several disciplines aimed at reducing the suffering of patients with neurological or psychiatric diseases: pharmacological, surgical, psychological, or behavioral approaches are available for the treatment of brain disorders at different levels. One of the approaches that has received much interest in the last decades is neurofeedback (NF) [3], a therapeutic approach based on operant conditioning applications to the electroencephalographic (EEG) activity control [4]. NF has proven to be effective in several disorders such as ADHD [5,6,7], insomnia [8,9], learning disabilities [10,11], migraine [12], depression [13,14,15], or anxiety [15,16]. However, for a number of subjects, the response to the NF intervention has not produced the expected results. Individuals undergoing NF have been classified as responders or non-responders [17] based on neurophysiological descriptors [18,19,20], psychological and neuropsychological variables [21,22], or socio-demographic factors [19,23]. Currently, empirical studies on one of the core features of operant conditioning are lacking, which has led some authors [24] to conclude that failure of NF could be motivated to some extent by methodological problems when implementing operant conditioning to brain waves.

### The Role of Reinforcers in Operant Conditioning

The theoretical basis of operant conditioning is that the behavior can be modulated by reinforcers, which are defined as a pleasurable consequence that follows after that behavior, e.g., providing food to a dog every time it performs a behavior that we want to promote [25].

Previous research has shown that several aspects of that reinforcer will determine its effects on the learning and the response behavior that is being conditioned [25]. Hutt [26] found that the growth in the reinforcer quality was followed by an increase in the response rate. He trained three groups of rats to press a bar using three different types of reinforcers: staple food, saccharin-enriched staple food, and citric acid-reduced quality staple food. Hutt observed that the response rate was higher in the group with the saccharin-enriched staple food compared to the group that received the staple food and to the other group that received the reduced-quality staple food. Those findings lead him to conclude that the conditioning efficacy is associated with the reinforcer’s capacity to appeal. Likewise, the subject’s expectations related to the reinforcer play a fundamental role in the behavioral learning. Hulse [27] stated that an increase in the expected quantity or the nature of the reinforcer produces a higher response rate. He conducted an experiment to condition three groups of rats to press a bar. One was trained with a constant 10 pellets reinforcement, another with a constant 5 pellets reinforcement and the last with a variable reinforcement that could be either one or 10 pellets. Later, three conditions were applied consecutively to each group: reinforcement with 1 pellet, with 5 pellets and with 10 pellets. His results indicated that, when the number of pellets dispensed was greater than what they expected from their previous training, the latencies to press the bar decreased even more.

Theoretically, this phenomenon is expected to be similar in the context of the NF. That is, a high-quality reinforcer should generate more positive results. However, previous NF works were designed in such a way that every participant received the same reinforcer. Besides, in most cases, the reinforcer value was more informational than motivational. This means that the stimuli employed to produce conditioning in NF could not have a value per se. For example, Hoedlmoser et al. [28], in a study to test the impact of sensorimotor rhythm (SMR) training with NF on sleep and declarative learning, used a needle moving to the right as a reinforcer. If the participants managed to surpass a certain amplitude of the SMR, they were reinforced with the image of a star. In this case, they achieved good results with an improvement in both learning and sleep. Schabus et al. [29] carried out a similar project in order to check NF efficacy in insomnia. These authors applied the same reinforcer to all their participants (i.e., an arrow that moved to the right when the subject performed in the correct direction) and found that the results for the trained group did not differ from those for the placebo group. The use of the same reward for all participants, without taking into consideration its value as a reinforcer for each individual, could lead to misinterpretations of the NF efficacy. Thus, what could be interpreted as a NF failure might be actually reflecting the absence of utility those particular stimuli have to modify behavior.

To our knowledge, the effects that the value of the reinforcers on the success of NF have never been tested. Since it has been reported that positive results are achieved when the subjects choose the reinforcers with individual and personal value for them [9,10], the aim of the present work was to analyze whether the effects of NF on brain waves differ when reinforcers are chosen by the subject. Having this information would be a major contribution to the field of NF, since the effects of NF that are limited in some subjects could be improved by selecting a relevant stimulus according to the individuals’ expectations and preferences.

## 2. Materials and Methods

### 2.1. Participants

This study was conducted in four different neurorehabilitation clinics in Spain (NEPSA Rehabilitación Neurológica—Salamanca and Valladolid-, Psyd Neurofeedback -Valencia- and Luria Rehabilitación Neurológica-Jerez, Cádiz-). Potential participants were included if they met the following criteria: (a) being aged between 18 and 65 years old; (b) not having a personal or family history of mental illness, brain injury, neurological disorder, serious medical condition, drug/alcohol addiction that could potentially impact cognitive functioning; (c) signing the informed consent; (d) not having lifetime experience with NF. Exclusion criteria were a) having suspected cognitive decline based on the Montreal Cognitive Assessment (MoCa) cut-off points; (b) having visual or hearing impairments despite correction. This study was approved by the institutional ethical committee (Nº3308 from 05.05.2020).

### 2.2. Measures

#### 2.2.1. EEG Collection and QEEG Analysis

An individual EEG recording session was performed on each participant before and after the NF session. Both the EEG recording and the NF intervention were conducted in the same session. To record the EEGs, 3 silver electrodes were used: the Cz was the selected 10/20 System point, the right earlobe being the ground and the left earlobe the reference. Both pre and post NF EEG signals from Cz were obtained and collected using an Atlantis-I amplifier from BrainMaster Technologies, Inc (Bedford, OH, USA). Impedances of less than 5K Ohms were maintained. We used an artefact rejection threshold of 100 µV, a 50 Hz notch filter and a sampling rate of 256 Hz. EEG records were recorded in open eyes condition using BrainMaster Technologies, Inc. Brain Avatar 4.6.4 software.

The QEEG values were analyzed using BrainAvatar 4.6.4 Review Session. Average SMR amplitudes [30] for pre and post NF EEG signals were computed as the average power in the 12–15 Hz frequency band over a 3 min window, by using digital filter.

#### 2.2.2. Montreal Cognitive Assessment (MoCA)

The MoCA is one of the most common tests for measuring global cognition and detecting cognitive impairment [31,32]. The MoCA has 30 items which could be categorized in several cognitive domains: executive functions, visuospatial function, short-term memory, language, attention, concentration, working memory, and temporal and spatial orientation [33,34]. MoCA administration takes approximately 10 min. Cognitive impairment is defined as raw scores lower than 26 according to normative data for Spanish population [34]. MoCA scores were interpreted according to age- and education-corrected scaled scores. Impaired performance was identified as scaled scores equal to or lower than 6.

#### 2.2.3. Visual Analogue Scale (VAS) for Reinforcer Rating

After the NF session, the subjects rated the reinforcer value using a subjective VAS that included a 10 cm line with numbers ranging from 0 to 10, with zero being no enjoyment at all and 10 being very high level of enjoyment.

### 2.3. Procedure

Participants were randomly assigned to one of two NF conditions according to the order of inclusion in the study. All the participants in the NF-selected group were offered games, music or films (Netflix or Amazon Prime) as a reinforcer. All the participants selected a movie, which differed among participants (Figure 1). The participants in the NF-imposed group were given the same videogame and, thus, were not allowed to select either the form or the type of feedback. The videogame consisted of a jar where there were a number of balls, and every time the criterium was met (put SMR voltage above the threshold) additional balls were introduced in the jar (Figure 2). In addition, we created a *NF-sham* group with the last 20 volunteers who were not aware of their placebo condition and who, as the selected group did, always chose the type and form of the reinforcer.

### 2.4. SMR Training Neurofeedback

We run a single neurofeedback session training SMR on Cz. To train SMR we used BrainAvatar 4.6.4 and Atlantis I amplifier (BrainMaster Technologies, Inc.). Usually, the clinician sets the amplitude thresholds so that irrespective of the selected frequency band, the software algorithm allows the patient to receive feedback. The goal is to ensure the reinforcement to the patient 50% of the time, although the literature specifies this percentage ranging between 20–70% [35]. We set up an auto-thresholding protocol that auto-adjusts the level of voltage to be achieved by SMR to get the feedback, the criterium to get the feedback was to put SMR voltage above the threshold, and the threshold was auto-adjusted to provide feedback 50% of the time. When more than one frequency band is being reinforced and/or inhibited, all set thresholds must be within the range that was set to receive feedback [36]. SMR amplitude was quantified by using digital filter and the update rate was 32 milliseconds on the feedback.

For the NF-selected group, a dimmer was placed in front of the video screen that offered sharpness when the patient met the criterium (put SMR voltage above the threshold) or became opaque, preventing the video from being viewed, when the criterium was not met (Figure 1).

The *NF-sham* group also was offered to choose both the form and type of the reinforcer, but we used an EEG simulation—a playback of a real EEG from a healthy individual—to run the session irrespective of the participants actual EEG.

### 2.5. Statistical Analyses

Categorical variables were compared between conditions using Chi-squared (χ^2^) test, whereas one-way Analysis of Variance (ANOVA) and independent *t*-tests were used to compare continuous variables.

A series of repeated measures ANOVA (RM-ANOVA) were used to analyze pre-post change in SMR. Both pre- and post-treatment SMR were checked for normality using Shapiro–Wilk’s tests. Because of the non-normality of the distribution of scores, both pre- and post-treatment SMR were log-transformed using the natural logarithm (ln). The first RM-ANOVA tested whether pre-post changes in the SMR differed between the groups (intervention vs. placebo). The second RM-ANOVA tested whether the pre-post change in SMR was associated with receiving either an imposed or a selected reinforcer within the intervention group. Group (0 = placebo, 1 = intervention), Condition (0 = imposed, 1 = selected), and sex (0 = male, 1 = female) were introduced as factors, whereas age was introduced as a covariate. The sphericity assumption was checked with Mauchly’s test.

The size of the effect associated to each variable (and interactions) in the repeated-measures ANOVA was assessed with partial eta2 (h2p) [37], with values of 0.01, 0.059 and 0.14 indicating small, medium and large effects respectively [37]. For mean differences, effect sizes were calculated with Cohen’s d, with values of 0.20, 0.50 and 0.80 indicating small, medium and large effects respectively [38]. Statistical significance was set at 0.05 for two-tailed tests.

## 3. Results

### 3.1. Differences between Groups

Data from 113 participants (intervention group *n* = 93, placebo = 20) were analyzed. Overall, there were 49 (43.36%) males and 64 (56.64%) females (Table 1). From the 113 enrolled participants, 48 were assigned to the *NF-selected* group and 45 participants were assigned to the *NF-imposed* group. There were no statistically significant differences between groups regarding level of education (χ^2^ = 3.82, *p* = 0.148) and sex ratio (χ^2^ = 0.03, *p* = 0.871). The group receiving placebo was younger [t(106.15) = −6.94, *p* < 0.001, Cohen’s d = 1.14] and had lower MoCA scores [t(42.64) = −2.37, *p* = 0.022, Cohen’s d = 0.49] than the intervention group (Table 2).

The RM-ANOVA showed that there were no effects of age [F(1,108) = 0.28, *p* = 0.595, η^2^*p* = 0.003], sex [F(1,108) = 0.30, *p* = 0.583, η^2^*p* = 0.003], time [F(1,108) = 1.52, *p* = 0.220, η^2^*p* = 0.014], or group [F(1,108) = 0.05, *p* = 0.830, η^2^*p* = 0.000] on the primary outcome. However, there was a statistically significant time by group interaction [F(1,108) = 4.94, *p* = 0.028, η^2^*p* = 0.044]. Paired-sample *t*-tests showed that log-transformed SMR post-treatment scores were similar to pre-treatment scores in the placebo group (paired *t*-test (*df* = 19) = 1.64, *p* = 0.118, Cohen’s d = 0.14), but higher than pre-treatment scores in the intervention group (paired *t*-test (*df* = 92) = −2.71, *p* = 0.008, Cohen’s d = 0.10).

### 3.2. Effects of Reinforcers on NF

There were no differences between *NF-imposed* and *NF-selected* groups in percentage of females (χ^2^ = 0.47, *p* = 0.491), age [*t*-score (*df* = 91) = 1.01, *p* = 0.317, Cohen’s d = 0.21], educational level (χ^2^ = 0.42, *p* = 0.812) or MoCA scores [*t*-score (*df* = 91) = 0.035, *p* = 0.972, Cohen’s d = 0.01] (Table 2).

Regarding the log-transformed SMR variables (Table 3), the repeated-measures ANOVA showed a statistically non-significant effect of time [F(1,88) = 2.59, *p* = 0.111, η^2^*p* = 0.029], sex [F(1,88) = 0.11, *p* = 0.744, η^2^*p* = 0.001], and condition [F(1,88) = 0.49, *p* = 0.486, η^2^*p* = 0.006]. However, a statistically significant Time by Condition interaction was found [F(1,88) = 4.39, *p* = 0.039, η^2^*p* = 0.048]. Paired-sample *t*-tests showed that log-transformed SMR post-treatment scores were similar to pre-treatment scores in the *NF-imposed* group (paired *t*-test (*df* = 44) = −0.52, *p* = 0.609, Cohen’s d = 0.08), but higher than pre-treatment scores in the *NF-selected* group (paired *t*-test (*df* = 47) = −3.03, *p* = 0.004, Cohen’s d = 0.44). The Mauchly test showed that the sphericity assumption of the repeated-measures ANOVA was not violated (*p* > 0.999).

### 3.3. Reinforcer Rating

Overall, the participants in the *NF-selected* group rated the reinforcer as significantly higher than participants in the *NF-imposed* group (*t*-test (*df* = 91) = 7.06, *p* < 0.001, Cohen’s d = 1.45). To analyze whether the pre-post difference in the SMR variable was related to the reinforcer rating scores rather than to selecting the reinforcer, we divided participants into two groups according to their reinforcer ratings. As the reinforcer rating scale ranged from 0 to 10, we labeled reinforcers as low for ratings 0–5, and as high for ratings 6–10. We then analyzed log-transformed SMR pre-post scores with a paired-samples non-parametric Wilcoxon’s rank test for each group within each condition. There were statistically significant differences in the percentage of participants scoring the reinforcer as high (χ^2^ = 29.92, *p* = 0.001). Within the *NF-imposed* group, there were no statistically significant pre-post differences in either the low (*n* = 29, 64.4%, *z* = 1.35, *p* = 0.177, Cohen’s d = 0.27) or the high reinforcer group (*n* = 16, 35.6%, *z* = 0.31, *p* = 0.756, Cohen’s d = 0.18). Within the *NF-selected* group, there were no statistically significant pre-post differences in the low reinforcer group (*n* = 8, 16.7%, *z* = 0.42, *p* = 0.674, Cohen’s d = 0.04). However, the high reinforcer group showed a statistically significant increase in post-test SMR scores compared to pre-test (*n* = 40, 83.3%, *z* = 2.98, *p* = 0.003, Cohen’s d = 0.50). As with the NF-*selected* group, most of the participants in the placebo group rated the feedback as high (*n* = 18, 90%).

## 4. Discussion

Since NF is based in operant conditioning of brain waves, it is important that the reinforcer is relevant for the subject [39]. In the present study, we aimed to analyze the effect the selection and the value of the reinforcer on NF has in healthy subjects. We found that participants who selected the reinforcer and, in addition, ranked it as high quality modified their SMR, a finding that was not replicated in participants who did not select the reinforcer. In addition, the superiority of NF over the sham group was also proven, since the placebo group failed to increase their SMR despite the fact that all participants in that group selected the reinforcer.

In our study, we found that participants who were not allowed to choose their reinforcement—the researcher chose it by default—did not show meaningful differences between their pre-session SMR and the post-session one. However, those who were able to choose their reinforcement showed a considerable increase in their post-session SMR. Moreover, those who could not choose their feedback did not show important differences between their post-session SMR and their pre-session SMR, regardless of how high or low they rated the reinforcer. This was also the case for the subjects from the group that could choose their reinforcements, and whose rating was low. However, the SMR for those subjects—who could choose their feedback and gave it a high score afterwards—increased significantly after a neurofeedback session.

The data suggest a double explanation for the results. On the one hand, the value of the reinforcer is key in the success of the EEG activity conditioning, and that is reflected by the fact that change was achieved by participants who have rated the feedback as very high. On the other hand, compared to the *NF-selected* group, participants in the *NF-imposed* group who rated the reinforcer as high did not improved their SMR after the NF intervention. If the increase in the SMR were exclusively related to the pleasantness of the reinforcer, it would be expected that participants in the *NF-imposed* group who rated the reinforcer as high should have shown improvements in the SMR similar to those shown by participants in *NF-selected* group. The lack of improvement in participants in the NF-imposed group who rated the reinforcer as high suggests that the key is not only the pleasantness of the reinforcer, but that it should also be relevant and motivational for the effect to be present. These results are in line with previous studies reporting that the reinforcer’s internal control locus [40], the motivation, and the expectations about the reinforcers [25,41,42,43,44] play a vital role. Neither the selection of the reinforcer (i.e., the internal control locus) nor its motivational value (translated as the relevance or the rates given to it by the subject) would explain in isolation the change in the selected group, since in the placebo group—where every subject chose their reinforcement—the reinforcer rating was having no effect on the SMR. Thus, two main findings must be highlighted: a) changes in SMR are a consequence of the NF training and are not expected in the absence of intervention, and b) changes in SMR are enhanced when the reinforcer is rated as highly enjoyable by participants receiving NF.

One of the reviewers of this work proposed an interesting alternative that could explain the increase in the SMR in the *NF-selected* group: this increase could be due to a novelty effect resulting in increased attention, which could in turn increase the SMR. This could occur during the NF session due to the novelty effect caused by the films, but not during the pre- and post-NF EEG recording—which was performed in a resting condition—since there is no stimulation of any kind. Some authors [45] have even found a spectral reduction, including the SMR, during attention tasks. However, it could be interesting to test whether the effect of the novelty produced by the reinforcer could modulate the increase of the SMR. Future studies should improve our design to control the effect of the novelty of the reinforcer on the SMR.

Some authors [24] reported that there is no evidence to prove the continuous feedback’s efficiency. In continuous feedback, the reinforcer is provided continuously and proportionally to the subject’s performance by inhibiting or reinforcing a brain activity. For example, in a car videogame, the better the performance of the subject is, the faster the car goes. They argue that a discrete feedback would achieve better results. Those authors support the previous studies vision [46,47] suggesting that the effects of the reinforcers on human conditioning are attributable to their informative value, not to their motivational function. They state their concern about the poor results found in some NF studies and attribute it to a potential lack of a correct methodology while applying the operant conditioning. In contrast, our results are in line with previous studies [39,48,49,50] reporting that the most relevant reinforcers for the subjects are those who help them learn more. For example, Fisher et al. [39] compared two methods to identify relevant reinforcers among 16 stimuli in people with serious disabilities. They later used the identified relevant reinforcers to modify the time subjects spent in a predefined place in the room. They found that the highly preferred stimuli were more effective in increasing the desired behavior. Mangum et al. [48] studied the efficacy of reinforcers in autism, dividing them in high and low preference reinforcers. For this purpose, they divided a room into several sides that were associated with different reinforcer conditions (high preference, low preference and absence of reinforcer). The target behavior was the entry and permanence of the subjects in each area. They found that high preference reinforcers were more effective in the pursue to attract subjects to the area where they were located. Svartal et al. [50] studied the value of the non-verbal reinforcer in humans and found that the motivational value of the reinforcer can be crucial in human conditioning as long as it is unrelated to verbal stimuli, as it is the case in NF. The authors created two groups that were given either a high motivational value reinforcer—lottery tickets that had a high monetary value and that also were exchangeable for prizes—or a low motivational value reinforcer—a small amount of money. The task consisted of differentiating between pairs of patterns, which were presented visually, and the behavior to be conditioned was the pressure they applied while pressing a button. They found that the high motivational value reinforcers resulted in a higher increase in the pressure made while pressing the button. Piazza et al., [49] in a study to assess the predictability of reinforcer efficacy, used three categories of reinforcers (high, medium and low preference) created by a task that the subjects had chosen. They subsequently evaluated each reinforcer based on the time the subjects spent at the location where each reinforcer was located. They found that high preference reinforcers were more effective than low and medium preference reinforcers in prompting longer intervals of behavior, this behavior being the reinforcer’s own usage. In addition, the motivational value of the reinforcer results in higher response rates [43,50]. Therefore, contrary to the suggestion made by Sherlin et al. [24], it seems that the feedback’s relevance is a key feature that should be taken into account in NF interventions.

Many studies on NF used either the same reinforcer (e.g., the same videogame) for all participants or just a signal indicating that the subject is meeting the criteria designated in the NF protocol. In the first case, it must be taken into account that the same reinforcement does not necessarily have the same effect on every subject; not even on the same subject over time [51]. In the second case, the signal indicating the subject how their performance is going would become a source of dichotomous information (the objective was achieved vs. it was not achieved). Therefore, the reinforcement would ultimately be taken from the emotional/motivational consequence that such information would provoke in them, but the signal itself does not represent a real reinforcement. The best solution seems to be a combination of both the right type of feedback and the right type of reinforcement for each subject (personalized) [52].

The success of NF is related to the effectiveness of the learning process [53]. Reinforcement is a critical factor in instrumental learning, which sometimes occurs while the subject is not aware of how change is developing or how it is being sustained [54,55].

Positive reinforcement seems to be more important than the feedback’s operating component [56]. Although some studies have found good results while using the same reinforcement for all participants [28,57,58], other studies have found limited effect or no effect at all [29]. Our results suggest that one cause for these discrepancies may reside in the use of the same or low-preference reinforcements for all participants, an issue that had not been monitored for so far.

A study limitation is that we used a design that includes a single NF session. However, previous studies [59,60,61] have used a similar design proving that it is possible to change the EEG activity only after a single session. For example, Nam and Choi [40] used the single-session SMR protocol to investigate whether the reward difficulty, according to the threshold settings, affects the changes in NF. They found important changes in the participants after just one session. In the same vein, in a sham-controlled study, Escolano et al. [59] discovered significant changes after a single-session to reinforce the Alpha activity. Lee et al. [60] showed that those changes were possible after one Alpha/Theta protocol session; they specifically found a significant increase in the Alpha activity. In a trial that involved patients with depression, MacDuffie et al. [61] used fMRI-NF to improve the mood regulation skills they had learnt in the cognitive behavioral therapy sessions, and found relevant clinical changes after a single session. This shows that changes in the EEG are possible after just one NF session. Even so, the generalization to practice is limited since most NF interventions always use a larger number of sessions [62]. Future research should focus on whether this same trend occurs in treatment protocols with a greater number of sessions and whether this approach, using high preference enhancers, could reduce the number of sessions of conventional treatments. Another limitation was related to the different reinforcer modalities. All participants in the NF-selected group received movie feedback, while the NF-imposed group were given the same videogame. Both, movies and the videogame, could have different effect on SMR training. However, Autenrieth et al. [63] have shown that reinforcers of different modality are equally effective as a medium to provide NF. These authors found no differences in the efficacy in increasing the SMR in a single session between two reinforcers of different modality. One consisted of bars that provided the subjects with information about their performance (informative value) and the other consisted of a videogame (with greater motivational value than the bars). Both reinforcers were imposed by the researchers.

In the present work, we aimed to fill a gap in the literature on the influence of the value that the reinforcer has for the subject in NF. From a clinical point of view, it represents a useful contribution for the optimization of the NF-based treatments. Future research areas should include other types of reinforcer, such as those that are solely for information purposes, and check whether the findings are similar in other reinforcer modalities. This study has shown the importance the motivational value of the reinforcer chosen by the individual has on their NF performance. Our results suggest that the value of the reinforcer is a key point for feedback efficacy and must be modeled when designing NF interventions.

## Figures and Tables

**Figure 1 brainsci-11-00457-f001:**
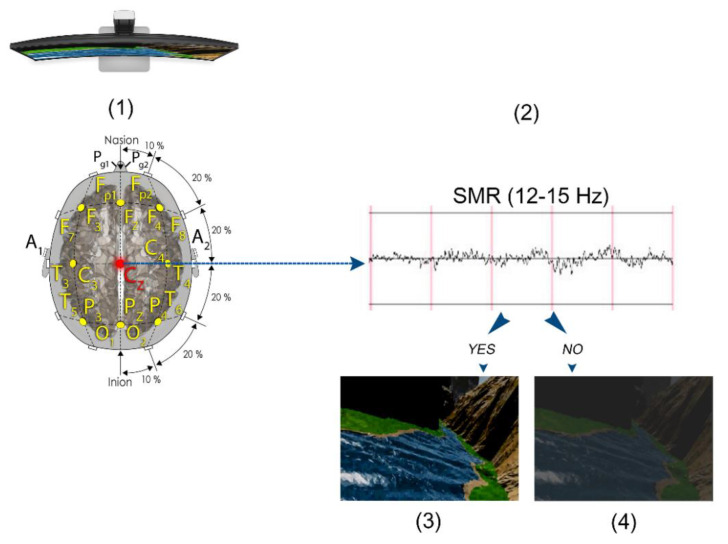
The subject is placed in front of a screen where the movie of their choice is projected, and their electroencephalography (EEG) activity is being recorded (**1**). Their sensorimotor rhythm (SMR) is calculated in real time (**2**). If their SMR goes above the configured threshold, then they get to see the image (**3**); if they do not get their SMR above the threshold or it falls below the threshold, then the screen dimmers preventing the subject from watching the movie (**4**).

**Figure 2 brainsci-11-00457-f002:**
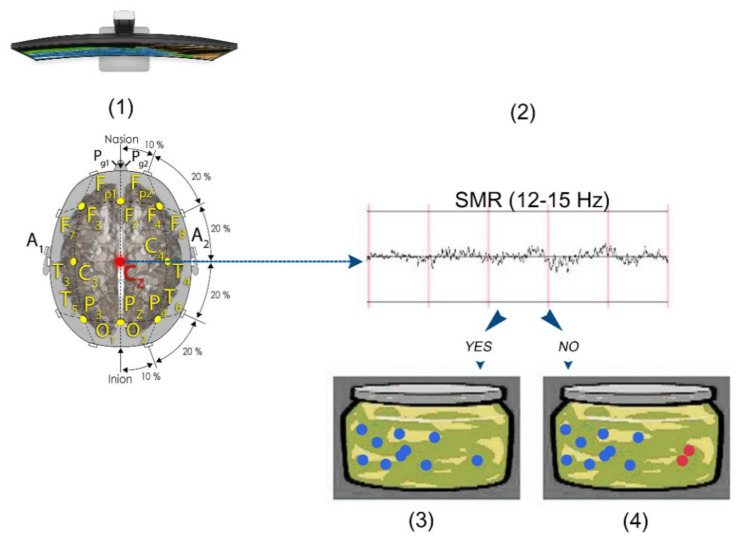
The subject is placed in front of a screen where the imposed videogame is projected, and their EEG activity is recorded (**1**). Their SMR is calculated in real time (**2**). If their SMR goes above the configured threshold, then new balls appear in the jar (**3**); if they do not get their SMR above the threshold or it falls below the threshold, then the new balls stop appearing and the ones they got before turn red and start disappearing (**4**).

**Table 1 brainsci-11-00457-t001:** Demographic variables.

		Imposed (*n* = 45)		Selected (*n* = 48)		Placebo (*n* = 20)		Total	
		* n *	%	* n *	%	* n *	%	* n *	%
Sex	Male	21	46.7	19	39.6	9	45	49	43.36
	Female	24	53.3	29	60.4	11	55	64	56.63
Education	Low	1	2.2	2	4.2	1	5	4	3.53
	Middle	10	22.2	9	18.8	8	40	27	23.89
	High	34	75.6	37	77.1	11	55	82	72.56

**Table 2 brainsci-11-00457-t002:** MoCA and Neurofeeback variables.

	Imposed (*n* = 45)		Selected (*n* = 48)		Placebo (*n* = 20)		Total	
	Mean	SD	Mean	SD	Mean	SD	Mean	SD
Age	38.42	12.17	35.90	12.02	27.05	3.25	35.38	11.61
MoCA	28.18	1.60	28.17	1.42	27.55	0.94	28.06	1.43
SMR Pre-treatment	4.54	1.64	4.53	1.40	4.51	0.83	4.53	1.41
SMR Post-treatment	4.60	1.77	4.79	1.44	4.41	0.89	4.64	1.50
Reinforcer rating	4.11	3.01	7.79	1.93	7.3	1.55	6.24	2.93

MoCA: Montreal Cognitive Assessment. SMR: sensory-motor rhythm.

**Table 3 brainsci-11-00457-t003:** SMR raw scores for each group.

	Imposed				Selected				Placebo			
	Low Feedback		High Feedback		Low Feedback		High Feedback		Low Feedback		High Feedback	
Variables	Mean	SD	Mean	SD	Mean	SD	Mean	SD	Mean	SD	Mean	SD
SMR Pre-treatment	4.50	1.84	4.61	1.25	4.55	1.32	4.52	1.44	4.99	0.15	4.45	0.86
SMR Post-treatment	4.64	2.01	4.53	1.27	4.61	1.45	4.83	1.46	5.05	0.24	4.33	0.91

SMR: sensory-motor rhythm.

## Data Availability

Anonymized data are available upon request.

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
