# Peer review of "Enhancing the Effects of Neurofeedback Training: The Motivational Value of the Reinforcers"

_brainsci, 2021, doi:10.3390/brainsci11040457_

Round 1

Reviewer 1 Report

 In this paper the authors describe the impact of the selection of reinforcers on the success of the NF training. By tuning the reinforcers for individual participants, the number of people who can benefit from NF therapy will significantly increase. 

Although the importance of this study for increasing the success of NF therapy, some changes will need to be made in the manuscript. 

  1.  What does pre and SMR and post SMR mean? This is not clearly defined in the methods. It is not told whether it will be performed on the same day or not. From the discussion I can conclude that it was the same day. But for me it was confusing while reading the manuscript.
  2. The materials and methods do not indicate how the distribution of the participants was made. You can therefore refer to Table 1 here. Were the participants in the placebo group deliberately selected for a lower MoCA score? 
  3. In Results it is stated that 94 of the 113 participants were assigned to the NF-selected group and 45 to the NF-imposed group (line 206). But together that is 139 (94+45). Something may have gone wrong here. 
  4. Table 1 is not clear. It is a combination of two tables. I would split this into two tables. 
  5. Sometimes Table is written with a capital letter in the text (line 211) and sometimes with a lowercase letter (line 228 and 229). This is inconsistent. 
  6. In figures 1 and 2, the legend does not properly describe what the numbers 3 and 4 mean. Number 3 represents a choice moment. The explanation of number 3 belong to number 4. Number 4 is not well described in the legend. It seems to me that number 4 should be number 5. 
  7. In figure 2 it seems that extra balls are created at number 5 (in red), but the text indicates that balls disappear here.  
  8. How can you predict with a single session that motivation will remain high? Would you suggest to change the reinforcement for each new session, or will it be determined over and over again during subsequent sessions?

Reviewer 2 Report

Review of manuscript: Enhancing the effects of Neurofeedback training: the motivational value of the reinforcers. This manuscript was submitted to the journal brain sciences.

Overall recommendation: except with minor revisions

Summary: the purpose of the paper was to show that if participants trained to increase SMR were able to choose their reinforcer they could significantly increase measures of this brain activity better than those that were not given the choice of reinforcer or controls who obtained random feedback. The abstract and discussion sections of the paper are excellent. On line 45 they use the term diseases to describe ADHD, insomnia, learning disabilities, and other disorders. These are not diseases and should be referred to as simply disorders.

Even for those individuals who chose as a reinforcer movies they were really only receiving partial reinforcement, technically variable ratio reinforcement. The reason for this is because SMR occurs in short spindles often lasting less than one second in duration. In order for the reinforcer to be accurate it has to track the duration of the SMR spindle. The authors do not describe how the SMR is detected, i.e. whether a multi-pole  analog, or digital filter was used to process the event when it took place. Since it is not possible to vary either the brightness, duration, or size of the movie display to match the short duration of the sensorimotor rhythm in effect they would be reinforcing both the occurrence of SMR as well as activity after the event which would match the duration of the movie. This leads to partial reinforcement. The author should describe in more detail exactly how the SMR event is processed in terms of its duration not just its amplitude. One could also propose that the increase in the SMR in the group using the movie could be due to a novelty effect resulting in increased attention which can also can increase SMR.

If novelty is an important factor it would have been better if they had employed up to five sessions at least to determine whether the differences between the two groups receiving reinforcement versus the controls continued over multiple sessions. They do mention in the discussion that they employed only a single session. It would be a contribution if in further studies they could look at the same design over multiple sessions with graphed data showing whether the Spearman rho correlation improves over time.

In summary, this reviewer recommends acceptance of the paper with minor modification describing in more detail the signal processing of the SMR events.

Reviewer 3 Report

The authors conducted a comprehensive neurofeedback study involving 113 subjects with control and sham feedback conditions, and highlighted a very interesting research question in neurofeedback, i.e., the motivational value of the reinforcers is important in neurofeedback success. The manuscript is quite good in general, however, in terms of the interpretation of the observed results, I think more details should be clarified and discussed at least.

  • Since the main question is to compare the difference between the NF-selected and NF-imposed groups. I would expect different reinforcers were selected by the subject in the selected group, and different reinforcers were randomly assigned to the subjects in the imposed group. By this way, the potential difference induced just by the different feedback modalities could be excluded. However, it seems in the current study, all 48 participants in the NF-selected group with real feedback selected the same reinforcer, i.e., movies, while all 45 participants in the NF-imposed group were received another movie feedback. By this way, to me, it is not very convincing to argue the difference between these two groups was contributed by the subjectivity of the feedback modalities, rather than just be dominated by the different feedback modalities.

  • It was nice to see the subgroup with high VAS values in the NF-selected group showed good NF effect. However, if the subjectivity on the reinforcer is really a significant factor on the NF effect, one would expect to see difference between subgroups with low and high VAS values in the imposed group as well. Because even in the imposed groups, if the subjects gave a high VAS values to the assigned feedback, then they would probably select this feedback when they have a choice. But it was not the case here, could the authors discuss why? Or alternatively, could is because the feedback provided to the imposed group was just less effective in general compared with the feedback in the selected group, which comes back to my first concern.

  • For the sham feedback group, please provide more details of how the EEG was simulated. How to guarantee that the simulated EEG has similar stationarity and variability, which would impact the received real-time feedback, compared with real physiological EEG? If not, for instance, if the received feedback in the sham group was much more unstable, flickering, and dynamic than the real feedback, then this could be a factor itself to fail the neurofeedback training effect.

  • In the abstract, it would be very helpful to clarify that the selected reinforcer indicates subject subjectively selected reinforcer, and the imposed reinforcer indicates the assigned reinforcer by the experimenter.

  • Is it true that all participants had the same frequency band for quantifying the SMR? If yes, what was it? If not, how to determine the frequency band for each subject? Also, what was exactly the SMR used for feedback, was it the time serials after band pass filtering at the selected frequency band, or the average power in the selected frequency band over a certain time window?

  • Page 6: I believe it should be 68 (48 real, 20 sham) in the NF-selected group, rather than 94. Please clarify.

Round 2

Reviewer 3 Report

I appreciate the authors’ effort in addressing my previous comments and I am also happy with most of them. There are still a few minor points which I hope could help to further improve the manuscript.

  • Since all participants in the NF-selected group received movie feedback (differed among participants though), while the NF-imposed group were given the same videogame, which was a very different modality compared with movie and might have different cognitive requirement or visual stimulation effect. I still don't think the authors could rule out the potential effect caused by the different modalities. A short discussion or limitation would be needed.

  • It is clear now that the SMR power for assessment the neurofeedback effect was quantified using average SMR amplitudes for pre and post NF EEG signals were computed as the average power in the 12-15 Hz frequency band over a 3-minute window, by using digital filter. My previous comment was actually asking the real-time processing of the EEG signal. Basically, to see the movie or control the videogame, the participants would need to increase the SMR voltage in real-time. The questions are how was the SMR voltage quantified in real-time? What was the update rate of this feedback signal? If the update rate was the same as the sampling rate (256 Hz), would it lead to a flickering, dynamic feedback which could be confusing for participants?

  • Page 3 line 126: “We used an artefact rejection threshold of 100µV, a 50Hz filter and a sampling rate of 256Hz.” Here I suppose 50Hz filter should be 50Hz notch filter.
